Best practices for assessing ocean health in multiple contexts using tailorable frameworks

Lowndes Julia S. Stewart 1 lowndes@nceas.ucsb.edu
Pacheco Erich J. 2
Best Benjamin D. 3
Scarborough Courtney 1
Longo Catherine 4
Katona Steven K. 2
Halpern Benjamin S. 1 5 6
1 National Center for Ecological Analysis and Synthesis, University of California at Santa Barbara , Santa Barbara, CA , United States
2 Moore Center for Science and Oceans, Conservation International , Arlington, VA , United States
3 Nicholas School of the Environment, Duke University , Durham, NC , United States
4 Department of Strategic Research, Marine Stewardship Council , London , United Kingdom
5 Bren School for Environmental Science and Management, University of California , Santa Barbara, CA , United States
6 Silwood Park Campus, Imperial College London , Ascot , United Kingdom
Toonen Robert
Electronic publication date: 2015 Dec 10
Publication date: 2015
Volume: 3
Electronic Location ID: e1503
Received 2015 Sep 15; Accepted 2015 Nov 24
Copyright: © 2015 Lowndes et al.
Copyright year: 2015
Copyright holder: Lowndes et al.
License: This is an open access article distributed under the terms of the Creative Commons Attribution License, which permits unrestricted use, distribution, reproduction and adaptation in any medium and for any purpose provided that it is properly attributed. For attribution, the original author(s), title, publication source (PeerJ) and either DOI or URL of the article must be cited.
License URL: https://creativecommons.org/licenses/by/4.0/

Keywords: Marine assessments, Ecosystem-based management, Open science

Funding: Pacific Life Foundation This work was possible with the generous funding from the Pacific Life Foundation to the Moore Center for Science and Oceans at Conservation International. The funders had no role in study design, data collection and analysis, decision to publish, or preparation of the manuscript.

==============================
Marine policy is increasingly calling for maintaining or restoring healthy oceans while human activities continue to intensify. Thus, successful prioritization and management of competing objectives requires a comprehensive assessment of the current state of the ocean. Unfortunately, assessment frameworks to define and quantify current ocean state are often site-specific, limited to a few ocean components, and difficult to reproduce in different geographies or even through time, limiting spatial or temporal comparisons as well as the potential for shared learning. Ideally, frameworks should be tailorable to accommodate use in disparate locations and contexts, removing the need to develop frameworks de novo and allowing efforts to focus on the assessments themselves to advise action. Here, we present some of our experiences using the Ocean Health Index (OHI) framework, a tailorable and repeatable approach that measures health of coupled human-ocean ecosystems in different contexts by accommodating differences in local environmental characteristics, cultural priorities, and information availability and quality. Since its development in 2012, eleven assessments using the OHI framework have been completed at global, national, and regional scales, four of which have been led by independent academic or government groups. We have found the following to be best practices for conducting assessments: Incorporate key characteristics and priorities into the assessment framework design before gathering information; Strategically define spatial boundaries to balance information availability and decision-making scales; Maintain the key characteristics and priorities of the assessment framework regardless of information limitations; and Document and share the assessment process, methods, and tools. These best practices are relevant to most ecosystem assessment processes, but also provide tangible guidance for assessments using the OHI framework. These recommendations also promote transparency around which decisions were made and why, reproducibility through access to detailed methods and computational code, repeatability via the ability to modify methods and computational code, and ease of communication to wide audiences, all of which are critical for any robust assessment process.

Introduction

Marine management is moving towards holistic ecosystem-based approaches with a coupled socio-ecological systems perspective. In light of this changing landscape, there is an increasing need to comprehensively assess coastal oceans by combining information from different sources, disciplines, and spatial and temporal scales to produce communicable outcomes (Levin et al., 2009; Mcleod et al., 2009; Tallis et al., 2010; Tett et al., 2013). Coupled human and natural systems assessments are most effective when they capture representative characteristics and priorities from different sectors and disciplines at scales relevant to management and policymaking (Borja et al., 2014). How this is achieved depends largely on availability of information, governance structure, and assessment time frame (Tallis et al., 2010).

The complexity of considering ecological, social, and economic variables simultaneously, combined with the need for high specificity and resource limitations, has largely resulted in ad hoc assessment frameworks being developed for each context individually. While such frameworks serve their purpose, they are context-specific and difficult to apply elsewhere, limiting the capacity of those in other institutions or geographies to build upon developed methods and capitalize on best practices. For example, integrated assessment methods have been developed at global (e.g., United Nations, 2014), regional (e.g., in the European Union: Marine Strategy Framework Directive, 2008), national (e.g., Australia: Parliament of Australia, 1998; United States: Millennium Ecosystem Assesment, 2005; National Ocean Council, 2013), and smaller scales (e.g., in the United States: National Marine Sanctuary, 2004; Puget Sound Partnership, 2008; West Coast Governors’ Agreement, 2006), each requiring significant time and effort to develop before use.

Ultimately, many if not most assessment processes lack transferability both in the lessons learned from developing the framework and in the utility of applying it to different contexts, which importantly includes using the same methods in the same location in future assessments to track changes through time. With the increasing need for ecosystem-based management, assessment frameworks should be developed for usability in multiple contexts, be flexible to improve upon past methods, and be able to accommodate differing geographies, ecosystem attributes, information availability and quality, cultural values, and political structures (Halpern et al., 2012; Samhouri et al., 2013). There must be higher levels of transparency (which decisions were made and why), reproducibility (access to detailed methods and computational code), repeatability (ability to modify methods and computational code), and ease of communication (for a wide audience) in assessment frameworks to enable progress through building from the lessons learned of previous approaches.

Here, we provide a unique perspective from experience with eleven completed marine assessments conducted at different spatial scales (global, national, and subnational) and contexts using the same assessment framework, the Ocean Health Index (OHI; Halpern et al., 2012). The OHI framework was developed conceptually and technically to be usable in different contexts, including in repeated assessments through time. We draw from our experiences conducting and supporting these eleven assessments and present broad best practices that are relevant to any assessment of coupled socio-ecological systems. Additionally, we provide tangible guidance and examples for conducting assessments using the OHI framework, and discuss how the framework has evolved and been used since its inception in 2012 (Halpern et al., 2012).

The Ocean Health Index

The OHI framework

The Ocean Health Index (OHI) is an assessment framework that comprehensively evaluates marine environments in a way that is standardized yet tailorable to different contexts and spatial scales (Halpern et al., 2012). OHI assessments combine open-source (freely-available) existing information and societal priorities to score marine systems according to the delivery of a suite of key societal ‘goals’ representing the benefits and services people expect healthy oceans to provide. OHI assessments can be conducted in different locations because the core part of the OHI framework (described in the next paragraph) is maintained while the tailored part of the framework allows flexibility regarding the goals that will be assessed and how they will be represented in a specific context. The OHI framework (Fig. 1) also includes the explicit definition that a healthy ocean is one that ‘sustainably delivers a range of benefits to people now and in the future’ (Halpern et al., 2012).

Figure 1 OHI assessments use the conceptual framework and the OHI Toolbox to calculate scores.

Both the framework and Toolbox have core parts that are standardized and familiar across all OHI assessments, while the tailored parts allow for customization to the local context. The framework is the conceptual design of the assessment, including the definition that a healthy ocean is one that sustainably delivers a range of benefits to people now and in the future. The core part of framework is that a suite of goals will be scored based on the current status, recent trends in status, and external pressures and resilience measures for each goal individually before combining goals. The tailored part of the framework identifies which goals should be assessed and how they should be represented as characteristics and priorities within the specific context. The conceptual framework is put into action using the OHI Toolbox, which is open-source, cross-platform, collaborative software built for calculating OHI scores. The ohicore R package is the core of the Toolbox; it is the engine to calculate OHI scores. The tailored part of the Toolbox provides structure for information processing and goal model development, and users have complete control over all decisions and analyses. Scores are displayed with figures, tables, and interactive maps on webpages available for each assessment. This figure illustrates just one phase of the assessment process, which cyclical and is followed by informing action and future assessments, learning from and sharing experiences, and planning improvements for future assessments to ultimately improve ocean health.

The OHI core framework has two fundamental attributes for calculating scores. First, it requires information on the status and trend for each goal and a wide range of pressures (negative influences) and resilience (positive influences) measures that will likely affect each goal status in the near term (Halpern et al., 2012). Second, input information and the goals themselves must have an explicit benchmark or ‘reference point’ to which they are compared (Samhouri et al., 2012). These reference points enable goals to be scored on a dimensionless scale from 0 to 100, where a score of 100 represents full delivery of the goal as defined by fully meeting the explicit benchmark. Reference points are set specifically to the local context to capture stakeholder priorities such that goal scores reflect progress towards achieving those targets, and future assessments can serve as a measure of the effectiveness of management and policy interventions.

Tailoring assessments to the local context includes identifying the suite of goals to be assessed, the unique suite of characteristics and priorities that should be captured within the goals, and how reference points should be set (Fig. 1). This process will help determine the information that should be used. Information included in the assessment will be guided and bound by the tailored framework and should be the best available information at the scale of assessment, (e.g., data, indicators, and knowledge of system functioning), the most appropriate assumptions to develop models to represent the goals, and the best-informed benchmarks. Because goal models are developed as a balance of ideal vs. practical (i.e., how the goal would ideally be represented in the local context vs. the information available and the understanding of system interactions), the models that represent OHI goals can often be improved at smaller scales. The OHI framework incorporates goals that have environmental and economic benefits but also ‘intangible’ ones that capture aesthetic, existence, and cultural values that are often excluded from tradeoff analyses. This opportunity to compare performance across disparate goals introduces challenges, since the dynamics of natural and socio-cultural systems are not fully understood, even in the most information-rich and well-studied areas, and societal preferences are not always agreed-upon, complicating quantifiable targets or rankings (Longo & Halpern, 2015).

Scores for each goal included in an OHI assessment are calculated for multiple Regions within the overall Assessment Area, and then scores for each Region are averaged (weighted by area) to obtain a single score for the entire Assessment Area. For example, if an Assessment Area was a country, Regions could be coastal provinces or states within that country. Scores are comparable among all Regions within an Assessment Area because all are calculated with the same methods. Calculating scores for each Region allows one to discover any spatial heterogeneity across the Assessment Area and identify geographic or thematic priorities for Regional management actions.

In the OHI framework, elements that contribute to the resilience of the system are those which humans can use to reduce the pressures exerted on oceans, thus increasing OHI scores. The OHI framework is designed so that the effectiveness of appropriate resilience measures (e.g., establishing protected areas, improving resource management regulations, improving governance, etc.) that decrease pressures and ultimately increase status scores can be tracked through repeated assessments using updated information from the same sources and the same models. This has been tested in assessments at the global scale (Halpern et al., 2015; ohi-science.org) but the true power of tracking change over time will occur at smaller spatial scales, where scores through time can be used to inform policy decisions, in part by surveying the effectiveness of management actions aimed at maximizing sustainable productivity while preserving vital ocean benefits.

While the framework can be used to assess changes in benefits across goals and for overall ocean health for a given management scenario, such as decreasing land-based pollution or increasing habitat restoration (Halpern et al., 2014), it is not a predictive tool, as is the case for other more discipline-specific methods that include mechanistic components with dynamic behavior (e.g., InVEST: Arkema et al., 2015). Uncertainty due to model specification and information inputs can be very difficult to estimate, but can be partly addressed via thorough and transparent documentation of all the parameter and modeling choices (M Frazier et al., 2015, unpublished data). In this sense, the OHI framework can be used to identify all instances where assumptions are being made, including where gap-filling or other input modifications occurred, which is an advantage over other integrated assessments that rely upon qualitative syntheses of information (Longo & Halpern, 2015).

Assessments using the OHI framework

Tailoring methods for any specific Assessment Area renders that assessment unique to its context, such that scores are only directly comparable between Regions of the Assessment Area and through time. Direct, quantitative comparison of scores with other Assessment Areas is limited, as assessment methods used in different locations may reflect different objectives or assumptions, for example by attributing more or less weight to different goals, thereby representing different visions of a healthy ocean and different expectations of what it should provide. Nevertheless, the use of a consistent framework for OHI assessments conducted in different Assessment Areas permits qualitative comparisons to be made, i.e., Assessment Area X is closer to achieving its stated management targets (reference points) than is Assessment Area Y. Further, as the framework is used multiple contexts with the same standard core structure, there is increasing familiarity of how to uncover the context-specific methods and approaches and interpret the details and results of the assessment process.

The ability of the OHI framework to accommodate different contexts has enabled assessments across many geographies, including in developing and developed countries, in information-limited and -rich areas and in relatively small and very large areas. To date, eleven OHI assessments have been completed, including four annual assessments of the world’s coastal exclusive economic zones (EEZs) (Halpern et al., 2012; Halpern et al., 2015; ohi-science.org), single-year assessments of Brazil (Elfes et al., 2014), the US West Coast (Halpern et al., 2014), and Fiji (Selig et al., 2015), and independently-led assessments (called ‘OHI+’) in Israel (Tsemel, Scheinin & Suari, 2014), Canada (R Daigle et al., 2014, unpublished data), Ecuador’s Gulf of Guayaquil (SETEMAR, 2015, unpublished data), and China (State Oceanic Administration, 2015, unpublished data) (Table 1). Others are also underway (the Arctic, the Baltic Sea, British Columbia, Chile, Colombia, Hawaʻi, Peru, and several locations within Spain).

Table 1 Information for completed and underway assessments using the Ocean Health Index (OHI) framework.

Scores are calculated for each spatially-defined Region within the overall Assessment Area. When assessments are conducted at national or sub-national scales, information from the global study can be used if no local information is available, since the Assessment Area of a national-scale assessment would be a Region of the global assessments. While this is possible, such information is not at high resolution and local-scale information is preferable to understand patterns within a national-scale assessment. Independent assessments (also called OHI+) are assessments led by independent groups (with primary project leads indicated in the ‘led by’ column) but with conceptual and technical support from the OHI team (including the authors of this study).

Assessment Area and year completed	Regions	Led by	Citation	Details	
Global 2012	171 nations, territories	Academic	Halpern et al. (2012)	First assessment using OHI framework. 10 goals assessed as defined in Table 2	
Global 2013	221 nations, territories	Academic	Halpern et al. (2015)	10 goals assessed. Improved models for tourism & recreation, food provision goals; redefined Regions	
Global 2014–2015	221 nations, territories + 15 high seas + Antarctica	Academic	ohi-science.org	10 goals assessed. Improved data processing methods and transparency	
Brazil 2013	17 states	Academic	Elfes et al. (2014)	10 goals assessed. Local models for tourism & recreation, artisanal opportunity goals; ∼20% local information used for goals, pressures and resilience	
Fiji 2013	1 nation	Academic	Selig et al. (2015)	10 goals assessed. Local models for food provision, artisanal fishing opportunities, sense of place, coastal protection goals, and habitats sub-goal. ∼10% local information used for goal status	
US West Coast 2013	5 states, substates	Academic	Halpern et al. (2014)	9 goals assessed. Local models for food provision, tourism & recreation, artisanal opportunity goals; excluded carbon storage goal; ∼80% local information used for goal status, pressures and resilience	
Canada 2014 (OHI+)	1 nation	Academic—Canadian Healthy Oceans Network	R Daigle et al., 2014, unpublished data	10 goals assessed. Redefined artisanal fishing opportunities goal as Aboriginal opportunities; public survey to determine goal weightings; ∼5% local data for goal status, pressures and resilience	
Israel’s Mediterranean Coast 2014 (OHI+)	6 districts	Academic + government— Hamaarag Israel’s Biodiversity Assessment Programme	Tsemel, Scheinin & Suari (2014)	First independent assessment using OHI framework and software. 9 goals assessed. Redefined natural products goal as desalinated water production. Local models for tourism & recreation, artisanal fishing opportunities, natural products; excluded carbon storage goal; ∼80% local data for goal status, pressures and resilience.	
Ecuador’s Gulf of Guayaquil 2015 (OHI+)	3 provinces, subprovinces	Government— Technical Secretariat of the Sea (SETEMAR)	SETEMAR , 2015, unpublished data	10 goals assessed. Local models for tourism & recreation; ∼82% local information used for goal status, pressures and resilience	
China 2015 (OHI+)	11 provinces	Government—China State Oceanic Administration	State Oceanic Administration, 2015, unpublished data	10 goals assessed. Local models for tourism & recreation, artisanal fishing opportunity; ∼71% local information used for goal status, pressures and resilience	
Arctic (OHI+)		Academic—Imperial College London	In progress	Emphasis on uncertainty in marine assessments	
Baltic (OHI+)	Combination of biogeographic and political regions	Academic—Stockholm Resilience Centre	In progress	9 goals will be assessed; multi-national effort	
British Columbia	8 subprovinces	Academic	In progress	Will track assess all goals for the past ten years and to explore how management may have contributed to these patterns	
Chile (OHI+)		Academic + government	In progress	Focus on the growing fisheries and mariculture sectors	
Colombia (OHI+)	3 national regions: Atlantic, Pacific, Caribbean Islands	Government—Colombian Oceans Commission	In progress	Assessment developed with 113 national indicators and through a multi-agency working group	
Spain’s Bay of Biscay (OHI+)		Academic—AZTI-Tecnalia	In progress	Assessment aimed at comparing findings with the results of other methodologies	
Spain’s Galicia (OHI+)		Academic— Campus do Mar	In progress	Assessment led by an academic-government-private sector consortium aimed at improving the sustainability of private sector activities and public sector decision-making	

Assessments conducted at smaller spatial scales often have higher quality information available and a better knowledge of system processes, which allows for goal models to be developed that better capture the philosophy of the goal in the local context. At the same time, reference points can be set with more refined knowledge of preferences or priorities of local people or governments at spatial scales where management decisions are made. The OHI framework can be used at these spatial scales to simulate management scenarios, identify policy priorities (spatially and thematically), and improve resource allocation and the cost-effectiveness of management.

Several governments have begun using the OHI as a marine planning tool even before scores have been calculated because tailoring the OHI framework initiated an inventory of any existing information and knowledge that could be useful for the assessment. Because such information spans disciplines, space and time, such an inventory can itself guide policy through an understanding of the current policy, information and knowledge landscape. For example, governments in China and Colombia have initiated efforts to improve data availability and standardization of data collection and interpretation for use in OHI+ assessments. In these countries, assessments will be incorporated directly into local planning and scores will be used to inform priorities for marine management. These programs have been government-initiated, and have developed at different rates. For example, China has completed its first national assessment and is planning to conduct repeated assessments at smaller spatial scales while simultaneously creating monitoring programs based on information gaps highlighted from the first assessment. Meanwhile, Colombia has developed a multi-year assessment plan and has prepared an inventory of all marine-related information and has formed a multi-sectorial collaboration to standardize meta-data in preparation for conducting their first national assessment.

Further, the OHI has been used as a platform for stakeholder engagement to identify priorities and perspectives. For example, the Gulf of Guayaquil assessment led several workshops with stakeholders representing coastal communities, fishing peoples, tourism industry, academia and conservationists to determine relevant goals, discuss sustainable management targets, and identify existing pressures on the local ecological and social systems. And to determine Canadian preferences for how goals should be weighted before being combined (with no information they are weighted equally by default), the Canada assessment conducted a national telephone survey that also can be analyzed for spatial and generational patterns (R Daigle et al., 2014, unpublished data).

OHI+ assessments are facilitated by a suite of open-source tools and instruction. The OHI Toolbox provides structure for data organization and storage, with data processing and goal modeling done in the programming language R (R Core Team, 2015) and R Studio (RStudio Team, 2015) for reproducibility and repeatability (Fig. 1). The OHI Toolbox is stored on the open-source online platform GitHub (GitHub, 2015), which allows for transparency and collaboration and also houses websites to display and communicate methods and results with interactive visualizations. More information can be found at ohi-science.org.

Best Practices for Integrated Assessments

Drawing on our experiences developing and supporting a wide range of OHI assessments at various spatial scales (Table 1), we present our findings as four best practices broadly applicable to any integrated assessment process. These best practices also include specific direction to guide assessments using the OHI framework and are accompanied by examples from completed OHI assessments. They are presented in sequence of occurrence in a given assessment process but should not be interpreted as enumerated steps because the process often involves iteration and revision, which occurs collaboratively and often out-of-sequence.

Incorporate key characteristics and priorities into the assessment framework design before gathering information

A conceptual assessment framework needs to simultaneously include broad ocean benefits and key characteristics reflecting social values and priorities specific to the context so the findings could be useful for established needs. These characteristics will extend beyond any single discipline or knowledge base and should be specified before information gathering begins. The importance of identifying these priorities prior to information gathering is relevant whether the framework is being developed de novo or whether an existing framework is being tailored for a specific location. When modifying existing frameworks, broad ocean benefits will likely already be identified, and tailoring the framework will involve confirming their relevance and incorporating key characteristics of the local context. The developed framework will provide the necessary structure to later identify, process and combine information. Doing this before assembling information is also critical because it can reduce potential bias, since an assessment framework built around the availability of known information may unduly skew the assessment and ultimately produce scores that misrepresent the assessment’s stated purpose.

Assessments using the OHI framework will be tailored to best represent the local context by identifying important local characteristics (both ecological and social) while still remaining true to the definition of the OHI (Fig. 1). The OHI defines a healthy ocean to be one that sustainably delivers a set of goals to people now and in the future, but what these goals are in specific detail and how their achievement is defined depends on the characteristics and stakeholder preferences of a specific Assessment Area. The ten goals included in global assessments (Table 2; Halpern et al., 2012; Halpern et al., 2015) have been identified as widely-shared across different geographies, but they are not universal, and assessments using the OHI framework should only include goals that are appropriate for the context (see Table 1).

Table 2 Table of ten goals and philosophies included in global Ocean Health Index assessments (Halpern et al., 2012; Halpern et al., 2015).

OHI goals have a two-letter code (example: FP, Food Provision) and any sub-goals have a three-letter code (example: MAR, Mariculture). Goals and sub-goals can be added, excluded or redefined, depending on characteristics of the Assessment Area.

Food Provision (FP)	The sustainable harvest of seafood in local waters from wild-caught fisheries (FIS) and mariculture (MAR; ocean-farmed seafood)	
Artisanal Fishing Opportunity (AO)	The opportunity for small-scale fishers to supply catch for their families, members of their local communities, or sell in local markets	
Natural Products (NP)	The amount of ocean-derived natural resources that are sustainably extracted from living marine resources	
Carbon Storage (CS)	The area and condition of coastal habitats that store and sequester atmospheric carbon	
Coastal Protection (CP)	The amount of protection provided by marine and coastal habitats serving as natural buffers against incoming waves	
Coastal Livelihoods and Economies (LE)	Coastal and ocean-dependent livelihoods (LIV; job quantity and quality) and economies (ECO; revenues) produced by marine sectors	
Tourism and Recreation (TR)	The value people have for experiencing and enjoying coastal areas through activities such as sailing, recreational fishing, beach-going	
Sense of Place (SP)	The protection of iconic species (ICO; e.g., salmon, whales) and geographic lasting special places (LSP; landmarks, ritual grounds) that contribute to cultural identity	
Clean Waters (CW)	The degree to which coastal waters are free of contaminants, such as chemicals, eutrophication, harmful algal blooms, disease pathogens, and trash	
Biodiversity (BD)	The conservation status of native marine species (SPP) and key habitats (HAB) that serve as a proxy for the suite of species that depend upon them	

Thus, how the OHI framework was tailored for the original global assessment (Halpern et al., 2012) and other completed assessments, should inform and guide, but not constrain, future assessments. When tailoring the OHI framework it is important to understand the rationale behind previously-developed approaches (Table 1) and to reflect whether any goals that should be added, removed or redefined. It is equally important to carefully consider and categorize how local pressures and resilience properties affect the goals and include them in the framework, making it informative and relevant to the local context. Tailoring the OHI framework is largely conceptual, and should be made before the technical details of data availability and modeling options are addressed to ensure that information availability does not inadvertently bias the assessment.

Examples: Several completed OHI assessments demonstrate how the definitions of individual goals were tailored within the OHI framework while maintaining its sentiment and definition of ocean health. In the Canada assessment, the perspective of Aboriginal communities, which has been absent from assessments at the global scale due to lack of information, was represented by redefining the artisanal fishing opportunities goal as the Aboriginal opportunity goal (R Daigle et al., 2014, unpublished data). In Israel, where the ocean provides half the drinking water for the country, the assessment framework was modified to capture the desalination of seawater, which was also absent from assessments at the global scale due to lack of information. It was included both as a component of the natural products goal and as a pressure on other goals (Tsemel, Scheinin & Suari, 2014).

Tailoring the OHI framework allows for goals to be omitted if they are not relevant to the local context. This was the case in the Israel assessment for the carbon storage goal (Tsemel, Scheinin & Suari, 2014), where there are no coastal habitats (e.g., mangroves, seagrasses or salt marshes) that contribute to carbon storage. Similarly, the natural products goal in the US West Coast assessment (Halpern et al., 2014) was omitted because harvest of natural products is not economically significant. In these cases, scores are calculated using only the remaining relevant goals. While omitting a few goals is permissible, narrowing the scope of the assessment too much reduces the capacity for comprehensive understanding of the interactions between ecological, social, and economic components.

Strategically define spatial boundaries to balance information availability and decision-making scales

Any assessment must explicitly define the spatial extent of the study area and any unique (non-overlapping) locations within, just as a management plan must define the management boundaries and a scientific study must define the area from which information is analyzed. Ideally, the spatial scale would be as small as possible to maintain the detail of the highest-resolution information available, allowing final results to be aggregated back up to coarser scales in explicit ways to match different management needs, and to ensure that all input information is being compared at appropriate scales. However, although spatially explicit information may be available at even finer scales, whether findings at such fine scale would be needed—or feasible across all benefits represented in the assessment—should be considered. Jurisdictional scales are often optimal, because information is typically collected in standardized and therefore comparable ways, and because this is the scale typically at which cultural priorities occur, management targets are set, and policy decisions are made. As a rule of thumb, assessment scales are most useful when they match the scale of decision-making.

For OHI assessments, identifying the spatial boundaries of the Regions within the Assessment Area is extremely important because OHI scores are calculated for each unique Region, and the boundaries will be used to aggregate or disaggregate input information reported at different spatial scales. Spatial boundaries should be defined with geographic information system (GIS) mapping software, ideally per management jurisdiction. Within the OHI framework, there is no limit to the number of Regions that can exist within the Assessment Area; the number is only constrained by data availability and the utility of having scores calculated for a particular Region. Although it is possible to assess only one Region in the Assessment Area (i.e., the Region is the Assessment Area, as in the Fiji assessment: Selig et al., 2015), this might not be ideal because it eliminates the possibility of making comparisons or identifying geographic priorities within the Assessment Area. When the Region is the Assessment Area the only comparisons that can be made are of the Assessment Area through time following subsequent assessments. Finally, while most assessments take place within political boundaries, it is possible to conduct assessments within ecological or physical spatial scales, such as basins, gulfs, seas or bays provided relevant information is available.

Examples: In the OHI assessment for the US West Coast (Halpern et al., 2014), finalizing the five Regions (Washington, Oregon, Northern California, Central California, and Southern California) was iterative as input information was assembled. Although information was often available at small municipal scales within each state, collection techniques differed among the three states (Washington, Oregon and California) and were therefore not comparable. Region boundaries were ultimately set as a balance between the available information that often came at the county-, or state-scale, and the fact that geographically California’s coastline was much longer and more diverse than is Washington’s or Oregon’s. As county-based information could be aggregated up and state-based information could be disaggregated down, it seemed appropriate to divide California into three Sub-Regions, with borders identified based on political (county) boundaries and biogeographic breaks.

Defining Regions in the Israel OHI+ assessment (Tsemel, Scheinin & Suari, 2014) also evolved during the assessment process. Israel’s Assessment Area was the Mediterranean coast and the initial plan was to assess the entire ∼270 km coastline as one Region. This seemed appropriate considering Israel is already a small area and there is value in reporting scores at the national scale. However, information was also available by district, which is the subnational jurisdiction level in Israel. Conducting the assessment instead with six districts allowed for more fine-scale information to be included in the assessment, simultaneously ensuring that these Regions were comparable in terms of information availability, type, and quality. A similar process occurred in the Gulf of Guayaquil assessment, where the Assessment Area was assessed as three regions instead of only one, as had been originally proposed (SETEMAR, 2015, unpublished data).

Maintain the key characteristics and priorities of the assessment framework regardless of information limitations

The assessment framework can be implemented using the best freely-available existing information, even if the information available is ‘limited’ or not ‘ideal.’ ‘Limited’ information may be of low quality, have gaps, or be indirectly obtained through modeling instead of being directly measured (see Tallis et al., 2010). Remaining true to the conceptual framework by incorporating indirect (proxy) or place-holder information, hence developing less-than-ideal goal models, provides a fuller picture than redesigning it to only include characteristics where ideal information is available. This is because all key characteristics in the system should be represented somehow in a comprehensive assessment, even if assumptions must be made to compensate for missing information. If these methods, including assumptions and rationales, are clearly considered and explained, completed assessments will not only provide the best possible picture of the current system but will also identify information gaps and highlight areas for improvement. Such scrutiny of available knowledge could be lost if important elements were simply excluded from the assessment due to imperfect representation.

It is widely recommended that assessments reflect the best understanding of local ecological, political, social, and economic processes, and information selection be closely guided by the purpose of the assessment (e.g., Levin et al., 2009; Tallis et al., 2010). Such information ranges from data, to indicators, to mechanistic understanding of the system’s functioning, to local priorities and preferences, and will come from many disciplines and sources. The selection of information necessarily involves a trade-off between the resources required for procuring, processing, or modeling it versus its relative contribution to the overall assessment. For transparency and repeatability, input information should preferably come from public or open data sources that are freely accessible and will continue to be updated regularly. These data should come from a trusted source, and directly measure a desired attribute at a high temporal and spatial resolution. These attributes are required not only for the completion and quality of the current assessment, but also to enable repeated assessments that are comparable because they use the same information sources, and are less resource-intensive because the data processing requirements are understood and scripted. Any assessment is only as ‘good’ as the information and decisions on which it is based, so using the full and most up-to-date array of information and knowledge sources available is fundamental to an integrated assessment. How to select information for the assessment (or exclude it) and how to fill any gaps is ultimately context-specific, and advice on how best to undertake this process has been presented elsewhere (e.g., Tallis et al., 2010; Halpern et al., 2015). Thresholds or benchmarks will be set according to the information used, and should be driven, whenever possible, by scientific knowledge and management preferences within the system (Tallis et al., 2010; Halpern et al., 2012; Samhouri et al., 2013).

Within the OHI framework, developing goal models and setting reference points is itself an iterative process, relying on the information available. The way goal models and reference points are finalized depends on what has been measured, with information searches extending beyond any one expertise, discipline, source, or data type. In many cases, ‘ideal’ information will not be available, requiring creative thinking about how to best use any proxy measures available, and will require additional assumptions regarding goal models and reference points. This process provides a useful means to identify information that can be easily integrated into the assessment, as well as information voids, which serves to prioritize future data collection.

Input information included in the assessment will require explicit, sensible reference points, as will the goals themselves. How these reference points are set should be founded in science and in management priorities, but they are often set subjectively in practice (see Samhouri et al., 2012). Stakeholder perspectives and literature review are highly valuable for setting reference points, and decisions will need to be made regarding the best appraisal of current information compared with targets. Ultimately, transparency in the decisions made is paramount, both for disclosing the methods used and for interpreting the scores (see next practice).

Examples: The OHI assessment in Brazil demonstrates how available information can be incorporated into previously-developed goal models while continuing to represent the sentiment of the goal. Trash that enters the ocean is a component of the OHI clean waters goal and is also a pressure on many other goals. In global assessments, models were developed based on available information reporting pounds of trash per mile removed from beaches for each coastal nation, with the assumption that garbage encountered on beaches is representative of the total amount at sea (Halpern et al., 2012; Halpern et al., 2015). To capture variation between Regions (coastal states) in the Brazil assessment (Elfes et al., 2014), information was required at a finer resolution. Quantity of trash removed from beaches was not available for each coastal state, but because the number of waste management services offered in each coastal municipality (access to beach cleanup services, household garbage collection, household recycling collection, and garbage collection in public streets) was known, they were aggregated and used as an indicator for each coastal state, with the assumption that more services meant less trash on beaches.

Global and US West Coast OHI assessments demonstrate how available information can be incorporated by developing new goal models while continuing to represent the sentiment of the goal. In the second global assessment (Halpern et al., 2015), a new model was developed for the tourism and recreation goal using the relative proportion of employment in the tourism industry rather than the number of international arrivals, their length of stay, and their population-weighted assignment to coastal areas, as had been done originally (Halpern et al., 2012). This new model was developed based on the assumption that employment statistics would be more closely correlated with coastal tourists than would international visitors, and also because it incorporated demand from domestic tourism. The US West Coast assessment, in contrast, developed a different model since more direct information about actual participation in nineteen coastal recreational activities was available (Halpern et al., 2014). This approach also set reference points based on each Region’s own temporal trends, eliminating the need to assume that these very different Regions should achieve similar rates of coastal tourism and recreation or attract and accommodate the same maximum number of tourists.

Document and share the assessment process, methods, and tools

Documentation of all aspects of an assessment process is paramount to ensuring transparency of the decisions made, reproducibility and interpretation of results, repeatability to facilitate and compare future assessments, and the ease of communication throughout the process. This means documenting and sharing not only the tools and methods used but also the knowledge gained through the process based on decisions made, what was decided against (e.g., why information was included or excluded, and how it was processed), challenges encountered, and recommendations for improvement. Frank documentation about the successes and shortcomings makes for greater scientific credibility, enables others to replicate what was done, and allows for the most appropriate interpretations of the results, as well as the highest potential for future improvement of assessment approaches, and ultimately, management towards ocean health. Providing public access to all such information (as both written text and programming code) is becoming the new standard for scientific inquiry (Hampton et al., 2014), so every effort should be made to achieve those aims.

Undoubtedly, methods can be improved upon in future iterations by incorporating new information or knowledge. Such improvements can come from within the original assessment team or advisors, for example by implementing approaches for which there had not been enough time or resources in the past, or from external feedback through academic literature, media, or other channels. Transparent and shared documentation can help both internal and external critique to be more informed and constructive, with different perspectives ideally leading to improvement in the ways marine systems are represented, understood, and managed.

The methods within completed OHI assessments have been incrementally improving with internal and external feedback. Feedback has been possible in part as a response to detailed documentation available for scrutiny, which itself has been improving with each assessment (including access to code and data). Internal and external criticism of the OHI framework (e.g., Branch, Hively & Ray, 2013) has largely centered on how it was tailored for assessments at the global scale: the quality of global data, the validity of goal models developed and reference points set with those data, and the relevancy of assessments at such a coarse scale. With this feedback, subsequent assessments have developed modified approaches with modeling several goals (e.g., wild-caught fisheries and tourism and recreation), setting reference points (e.g., mariculture), and tracking and highlighting information gaps (Halpern et al., 2015). Further improvements will be possible as small-scale and independent OHI assessments refine methods that can be incorporated into global assessments.

Example: To share what we have learned from conducting OHI assessments in addition to explicit methods, we have developed freely downloadable software tools and instructional materials describing the process. The ‘OHI Toolbox’ is built with the open-source and cross-platform R programming language (R Core Team, 2015), RStudio interface (RStudio Team, 2015) and the collaborative versioning platform GitHub (GitHub, 2015), tools used more and more by scientists (Dabbish et al., 2012; Ram, 2013). To assist independent groups now leading their own OHI assessments, tailored repositories containing template data and goal model scripts developed for global assessments are provided, and interested parties are given full control to update the input information, goal formulas, and reference points. Scores are calculated and displayed as tables, figures and interactive maps with the core part of the Toolbox, and shared online (Fig. 1). Improvements are ongoing and materials describing how to use these tools, as well as links to completed and on-going assessments, are available at ohi-science.org.

Discussion

Following many government mandates (e.g., Marine Strategy Framework Directive, 2008; The White House, Office of the Press Secretary, 2010) and a recent papal encyclical (Francis, 2015), there is greater impetus than ever to holistically understand and assess socio-ecological systems and design policy to improve the sustainability of human interactions with nature. Assessment methods to distill and communicate such complexity are imperfect but progressing (Halpern et al., 2012; Levin et al., 2013; Borja et al., 2014), and frameworks that are flexible to different contexts enable others simultaneously to build from past experiences and to tailor methods to local needs. Rather than developing location-specific frameworks de novo, the more that assessments can build directly from the successes and shortcomings of previous approaches, the more time and resources can be directed towards monitoring, scenario development, and management action. Further, assessments using a familiar framework help make methods and results more easily interpreted and communicated. Although input and model details will differ between assessments that limit how directly results can be compared, the familiarity of the framework allows for those details to be uncovered more easily.

To be most effective as a science and policy tool, assessment frameworks should be used repeatedly to track changes through time and evaluate the success of past and existing policies and inform future directives. This requires that assessment methods be transparent, reproducible, repeatable, and easily communicated to different audiences, and that they be flexible to improving knowledge, evolving priorities and newly available information.

The Ocean Health Index (OHI, Halpern et al., 2012) is one of several frameworks developed to date for assessment of coupled human and natural systems (Borja et al., 2014; A Borja et al., 2015, unpublished data), and is able to accommodate different contexts due to the structure and accessibility of the conceptual framework and of the Toolbox software (Fig. 1). The OHI framework has been used in assessments across different geographies and contexts following the four best practices described above (Table 1). While we have distilled our experiences leading and supporting assessments into those best practices, we also have other observations specific to the OHI framework that we share here:

• The OHI framework can be used as a policy tool even before scores are calculated. The OHI framework can be used to inventory marine-related information available for use in an assessment. This can identify priorities regarding knowledge and information gaps where indirect (proxy) information would be used in the absence of more direct measures, which can be useful well beyond OHI purposes. In Colombia, efforts to inventory available information have led to developing a central data repository and standardized monitoring program before the OHI assessment has truly begun.

• Scores are not directly comparable between assessments, but that is not their purpose.The purpose for conducting an OHI assessment is to best represent the local context being assessed, incorporating different components of what constitutes a healthy ocean, and tracking changes through time. Thus, the motivation is to compare spatially and temporally within the Assessment Area by employing uniform methods and reference points, not to compare to other locations outside of the Assessment Area. While methods tailored to a specific assessment make the resulting scores not directly comparable to scores calculated in any other Assessment Area, qualitative comparisons can be made because all OHI assessments share the standard core framework where resulting scores range from 0 to 100, with 100 representing what a healthy ocean should provide within any context. In this sense, there can be value in comparing between assessments and exploring how well stated management targets are being achieved in different Assessment Areas. The familiar structure (both in the conceptual framework and in the Toolbox) facilitates uncovering how goals were modeled and targets were set. While comparing across Assessment Areas would not be possible if the assessments used entirely different assessment approaches, it can be explicitly unveiled and communicated with assessments using the OHI framework.

• It is best to assemble a small core team of skilled, flexible people. Because OHI assessments combine knowledge from multiple disciplines, require developing models and setting reference points, and use innovative, evolving open-source tools, team members will likely be required to work outside of their direct expertise and technical ability and must feel comfortable and have the authority to do so. One approach that has worked well has been to assemble a core team that is very knowledgeable about the OHI framework and process, with specific members having backgrounds in marine science and tasked with information gathering and communication or with information processing and the Toolbox (or better yet, a combination). This core team then engages additional ‘goal keepers’ that are experts in specific OHI goals to help develop the best models, access information, and set reference points, through in-person meetings, workshops, and remote communication.

• Certain goals should be approached differently than they were in global assessments. Goal models developed for global assessments are limited by information available for all countries worldwide (although the ideal intent of the goal are set within the tailored framework); those global models should not limit the approaches used by smaller-scale OHI assessments. Instead, the tailored part of the framework should be designed with the best knowledge information and for the local context, particularly reference points.

• Resilience measures should be directly matched to specific pressures on the system. In the OHI framework, resilience measures represent how the pressures exerted on the marine system might be mitigated. Thus for repeated OHI assessments to detect improvement to ocean health, pressures should ideally be responsive to the resilience measures selected, which will serve to decrease the magnitude of those pressures, increase OHI scores, and ultimately ocean health.

• It is critical to complete the conceptual framework before using the OHI Toolbox. While the OHI Toolbox organizes input information and calculates scores (Fig. 1), its utility comes after much planning and preparation, after the tailored framework has been developed, spatial regions have been defined, and information has been gathered and confirmed to be appropriate. It is inefficient to gather and format data that is ultimately excluded from the assessment. The Toolbox and supporting resources will greatly aid the assessment process, but only after a large amount of work has already been done.

The OHI core framework, i.e., that goals are scored individually based on current status, trend, and external pressures and resilience measures before being combined together (Fig. 1), has not changed since its inception in 2012. However, any assessment methodology benefits from constant evaluation and improvement aimed at incorporating new knowledge, data and understanding. For example, we continue to work to improve technical aspects with the Toolbox, including how information quality and uncertainty is traced throughout the assessment. Further, we make improvements to the tailored part of the framework in annual global assessments by updating data sources as new options become available. For example, goal models for wild-caught fisheries and tourism and recreation have been redeveloped to accommodate new information and understanding, reference points for mariculture have been fine-tuned, and data processing workflows have been more standardized for natural products and species (Halpern et al., 2015). Additional improvements will come from OHI+ assessments that use local information and decisions to develop innovative approaches to represent complexity with small-scale systems using locally relevant reference points. It is exciting that this has already been the case in many geographies (see Table 1).

We have found through our experience with integrative assessments, specifically with the Ocean Health Index, that the most valuable lesson learned is that the process of conducting an assessment can be as valuable as the resulting scores. This is because an assessment process becomes a platform for collaboration and information sharing across perspectives, and can build foundations and catalyze progress well beyond any single assessment. Resources and workflows illustrating processes for conducting assessments are becoming more freely available (e.g., msp.naturalcapitalproject.org, ohi-science.org), and within that landscape we hope the best practices presented here will help guide future assessments and improved ocean health.

We would like to thank all past contributors and funders that made completed Ocean Health Index assessments possible, as well as groups that make information and tools freely available for use. We wish to give special thanks to Melanie Frazier, Casey O’Hara, Ning Mendes, Johanna Polsenberg and Lindsay Mosher, and PeerJ editor Rob Toonen and two anonymous reviewers whose comments greatly improved this manuscript.

Additional Information and Declarations

Competing Interests

Author Contributions

Data Availability

The authors declare there are no competing interests.

Julia S. Stewart Lowndes, Erich J. Pacheco, Benjamin D. Best, Courtney Scarborough, Catherine Longo, Steven K. Katona and Benjamin S. Halpern conceived and designed the experiments, performed the experiments, analyzed the data, contributed reagents/materials/analysis tools, wrote the paper, prepared figures and/or tables, reviewed drafts of the paper.

The following information was supplied regarding data availability:

Data repositories are housed at github.com/ohi-science and information and software training materials are found at ohi-science.org.

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
