# Peer review of "Best practices for assessing ocean health in multiple contexts using tailorable frameworks"

_PeerJ, doi:10.7717/peerj.1503_

## Round 0.1 · original submission · Minor Revisions

I now have 2 reviews back from your submitted manuscript, and while both are encouraging about the value of the submission, both also feel that the presentation could be improved. The referees are split on how serious a revision they would like to see, and I can see both sides myself, but both have a number of what I believe to be good suggestions for you in revising your manuscript. For example, there are a number of comments from both about the missed opportunities with this paper and the relatively soft presentation of the rules of thumb that you may or may not agree with. Regardless, I have to point out that these referees are your intended audience, and each has provided you with honest feedback about their impression of the paper. I have split the middle on the recommendations from the reviewers and request that you revise your manuscript in response to the referee suggestions, which I expect should help you to produce a better final version. I look forward to seeing your revised manuscript.

Reviewer 1 ·

Basic reporting

See comments below - this is not a typical research article, but rather a synthesis

Experimental design

See comments below - this is not a typical research article, but rather a synthesis

Validity of the findings

See comments below - this is not a typical research article, but rather a synthesis

Additional comments

This paper uses ten applications of the Ocean Health Index to summarize some best practices, namely that core values/characteristics should be agreed upon at the outset and maintained despite data limitations; spatial boundaries should be strategically defined at the smallest scale possible; and information should be shared.

Overall, the paper was well written, but it left me wanting more. While the distillation of best practices is fairly interesting and the recommendations certainly seem like common sense and may be useful for those embarking on their own OHI, there is nothing really novel about them. I expected a “so what” or “here’s where we tried to implement our best practices and this is the results”.

I think one missed opportunity of a paper that reviews the efforts done to-date to compile OHIs around the world is providing suggestions to improve fundamental OHI principles, process (e.g., for establishing goals --potential pitfalls/biases), etc. Are there lessons from these ten experiences? Now that we have 10 OHIs at different scales, are there any insights about how well it works across scales? Somewhat by necessity as the authors are also the group who supported most of the OHI exercises reviewed, the reference list is mainly self-citing. Other resources may help reflect on strength/weaknesses of OHI relative to alternative indicators or approaches, see lines 148-160, e.g.,
Tett, Paul, et al. "Framework for understanding marine ecosystem health." Mar. Ecol. Prog. Ser 494 (2013): 1-27. Similarly, are there lessons from other indicator exercises that provide insight?

One way that the impact of this paper could be increased would be to discuss how the information was used, and what the policy resonance was. Are there any lessons about how to embed OHI in policy making (other than the generic match the objectives, conduct an open process, etc.), or how to transfer responsibility for doing the OHI over time to public agencies to ensure its longevity? (It would be useful to describe who led the effort (academics, government, etc.) in each case.)

Regarding the best practice “spatial boundaries should be strategically defined at the smallest scale possible” – I found the wording a bit misleading. I do not think that you actually advocate using the finest scale – of course your use of the words “strategic” and “possible” are meant to greatly qualify “finest” but what you describe is a much more nuanced and iterative process of defining spatial scales/regions that balances the tension between the scale of available data, scales where you have no/limited data, scales where you can fill zero data, and decision-making scales. I would reword this best practice to reflect that. You might even reference literature on governance scales.

While you advocate making everything public, you don’t discuss the benefits of basing the indicator on public, open data or data that you know will continue to be collected over time – it seems both of these are really important for the sustainability of the exercise.

Minor comments:
p6 137 “resilience actions” – jargon?
278 delete extra “and”
149 missing an “are”
Para 299-312 a bit confusing, eg 209-310 you already made the point in the sentence above that it made sense to divide California into 3

Reviewer 2 ·

Basic reporting

This manuscript is a distillation of lessons learned from applying the Ocean Health Index in multiple contexts. As such, it is a useful document, not only for future implementations of OHI (which increasingly may be done independently, hence the value of these published findings), but also for implementation of other integrated assessments – as the authors point out themselves. It is this value beyond just the OHI, and the attention that the authors have paid to documenting and generalizing four key lessons, that make it useful as a published open-access document.

The OHI has face many technical criticisms over the years, many of which have resulted in improvements to the index, and in particular the view expressed in this paper, that multiple implementations in Assessment Areas, that may not be comparable with other ones, may be the most credible/useful way to move forward. This manuscript does not deal with those critiques, which are addressed methodologically in other places – this manuscript presents the overarching lessons that include dealing with these technical issues. In some ways it is quite ‘soft’ in presenting four rules of thumb/lessons learned, but this also makes it very accessible to a broad audience, including the many non-technical leaders for whom implementing an OHI may provide a very useful tool.

Experimental design

As noted above, the paper distils findings from 10 applications of the Ocean Health Index. The design of the paper is appropriate for reporting on these.

Validity of the findings

The findings are valid, and as noted above, useful for readers who are going to apply the OHI, or another integrated assessment. They are a useful contribution to the general literature assessing assessment approaches.

Additional comments

Some minor editorial items below, by line number

129 – this goes into using ‘region’ and ‘area’ in ways specific to OHI, and to my mind ‘regions’ would be larger than ‘areas’. To minimize confusion, it might be clearer to write “scores for each goal area calculate for MULTIPLE Regions within the OVERALL Assessment Area …”

137 – again, “Resilience actions … only ways …” this is a statement of the philosophy of OHI, rather than a completely factual statement. People can reduce pressures in multiple ways that might not be defined as ‘resilience actions’ by all.Changing the sentence to something like “In the OHI framework, resilience actions are key …”

149 – that ARE often excluded

211 “location” – isn’t it better to be specific to OHI terminology and replace this with Assessment Area

512 – Polsenberg mis-spelled!

---

## Round 0.2 · accepted · Accept

Thanks for your revision and your effort in addressing the comments of the referees. I believe that you have done a good job with your revision, and the new section addressing the assessment planning, discussing criticisms of the approach and planned improvements for the future is particularly appreciated. Overall, I feel that you have addressed the concerns of the referees to my satisfaction with the revision, and I am happy to move it along into production.